# Graphene Oxide Nanoplatforms to Enhance Cisplatin-Based Drug Delivery in Anticancer Therapy

**DOI:** 10.3390/nano12142372

**Published:** 2022-07-11

**Authors:** Elena Giusto, Ludmila Žárská, Darren Fergal Beirne, Arianna Rossi, Giada Bassi, Andrea Ruffini, Monica Montesi, Diego Montagner, Vaclav Ranc, Silvia Panseri

**Affiliations:** 1Institute of Science and Technology for Ceramics–National Research Council (CNR), 48018 Faenza (RA), Italy; elena.giusto@aol.com (E.G.); arianna.rossi@istec.cnr.it (A.R.); giada.bassi@istec.cnr.it (G.B.); andrea.ruffini@istec.cnr.it (A.R.); monica.montesi@istec.cnr.it (M.M.); 2Regional Centre of Advanced Technologies and Materials, Czech Advanced Technology and Research Institute, Palacký University Olomouc, 78371 Olomouc, Czech Republic; ludmila.zarska@centrum.cz; 3Department of Chemistry, Maynooth University, Maynooth, Ireland; darren.beirne.2017@mumail.com; 4Department of Chemical, Biological, Pharmaceutical and Environmental Sciences, University of Studies of Messina, 98100 Messina (ME), Italy; 5Department of Neuroscience, Imaging and Clinical Sciences, University of Studies G. d’Annunzio Chieti-Pescara, 66100 Chieti (CH), Italy; 6Institute of Molecular and Translation Medicine, Faculty of Medicine and Dentistry, Palacký University in Olomouc, Hnevotinska 5, 77900 Olomouc, Czech Republic

**Keywords:** nanomaterials, platinum-based drug, graphene oxide, breast cancer, osteosarcoma, glioblastoma, nanomedicine, drug delivery systems

## Abstract

Chemotherapeutics such as platinum-based drugs are commonly used to treat several cancer types, but unfortunately, their use is limited by several side effects, such as high degradation of the drug before entering the cells, off-target organ toxicity and development of drug resistance. An interesting strategy to overcome such limitations is the development of nanocarriers that could enhance cellular accumulation in target cells in addition to decreasing associated drug toxicity in normal cells. Here, we aim to prepare and characterize a graphene-oxide-based 2D nanoplatform functionalised using highly branched, eight-arm polyethylene-glycol, which, owing to its high number of available functional groups, offers considerable loading capacity over its linear modalities and represents a highly potent nanodelivery platform as a versatile system in cancer therapy. The obtained results show that the GO@PEG carrier allows for the use of lower amounts of Pt drug compared to a Pt-free complex while achieving similar effects. The nanoplatform accomplishes very good cellular proliferation inhibition in osteosarcoma, which is strictly related to increased cellular uptake. This enhanced cellular internalization is also observed in glioblastoma, although it is less pronounced due to differences in metabolism compared to osteosarcoma. The proposed GO@PEG nanoplatform is also promising for the inhibition of migration, especially in highly invasive breast carcinoma (i.e., MDA-MB-231 cell line), neutralizing the metastatic process. The GO@PEG nanoplatform thus represents an interesting tool in cancer treatment that can be specifically tailored to target different cancers.

## 1. Introduction

Cancer is one of the leading causes of death worldwide [1,2], and it is frequently caused by many factors, including genetic variables, physical and chemical insults and lifestyle habits, such as poor diet or smoking [3,4]. It has been widely studied based on its epidemiology and mechanism, with the main therapies adopted to date based on various surgeries, chemotherapy and radiotherapy, as well as relatively new strategies, such as immune or gene therapies. Following surgical removal, chemotherapy is the most common treatment strategy, as it is based on the use of specific molecules targeting the high cancer cell proliferation rate, interfering with their metabolism [5]. Among chemotherapeutic agents, platinum(II)-based compounds are the most successful, and three compounds, i.e., cisplatin, carboplatin and oxaliplatin, are used worldwide to cure several cancer types (Figure 1) [6,7].

The main target of platinum-based drugs is nuclear DNA; upon penetration of the cell membrane, cisplatin infiltrates the hydrolysis pathways forming cationic aqua species that reach and covalently bind to the nuclear DNA, causing its damage, as well as the arrest of the cancer cell cycle in the G2/M transition phase, leading to apoptosis [7]. Although platinum-based chemotherapeutics are the most frequently used anticancer drugs, they still have several side effects. Platinum is administered intravenously, abd the targeted cancer cells are reached the via blood stream, where the drug is usually bound to plasma proteins, such as albumin, and degraded in a high percentage before entering the target cells [7,8]. Systemic administration potentially leads to higher toxicity for off-target organs, eventually causing adverse side effects, such as nephrotoxicity and ototoxicity [9,10]. Moreover, tumour cell resistance to platinum has adverse and serious consequences for the fate of patients [11]. Ab interesting strategy to overcome these limitations is the development of nanocarriers that could enhance cellular accumulation, decreasing the associated toxicity [10,11,12]. Graphene oxide (GO)-based nanoplatforms have interesting physicochemical and surface properties, making them potentially attractive for medical applications as nanomaterials for cancer-targeted drug delivery [13,14]. However, water-dispersible GO often aggregates under physiological conditions in the presence of salts due to the charge screening effect [15]. In addition, depending on the concentration used, GO can per se induce cytotoxicity due to oxidative stress [16,17]. To target these problems, GO readily undergoes bipartite surface modification thanks to carboxyl groups abundantly present on its surface that allow for functionalisation with many biomolecules and drugs [18]. In order to reduce the level of cytotoxicity while promoting its cellular uptake, the use of bio-mimicking molecules, such as polyethylene glycol (PEG), has been recently studied [18], and it has been shown that the addition of PEG increases stability, improves solubility and reduces aggregation, prolonging the circulation of GO in the bloodstream [19,20]. 

In the present study, we focused on the preparation and characterization of a GO-based nanoplatform functionalised using highly branched PEG modality, namely 8-arm PEG, to enhance the efficacy and loading capacity of Pt-based drugs, thereby achieving a highly performant and stable nanodelivery system. Physicochemical characterization was carried out to investigate the features of the functionalised nanomaterial in terms of its dimensions, platinum drug-loading capacity and long-term stability. Its biological activity, such as cellular uptake, viability, morphology and migration, was then evaluated in seven human tumour cell lines selected as representative of three cancers with high incidence and morbidity worldwide, i.e., breast cancer: MDA-MB-231 and MDA-MB-486 cell lines; glioblastoma: U87 and U118 cell lines; and osteosarcoma: MG63, U2-OS and SAOS-2 cell lines [12,21,22].

## 2. Materials and Methods

### 2.1. Synthesis of Pt-Based Drug 1 (Pt-Free)

Compound **1** was synthesized as previously reported with few modifications (Figure 2) [23]. Oxoplatin, c,c,t-[Pt(NH_3_)_2_Cl_2_(OH)_2_] (0.1 g, 0.3 mmol) was suspended in 10 mL of DMSO. Succinic anhydride (0.028 g, 0.28 mmol) was added to the mixture, and the suspension was stirred at 45 °C overnight. The obtained solution was filtered to remove the small amount of unreacted oxoplatin and lyophilized overnight. The residue was washed with cold acetone, cold methanol and diethyl ether and dried under a vacuum (0.058 g, yield 62%). ^1^H-NMR (DMSO-d_6_) 6.50 (br.tr, 6H), 2.95–1.93 (m, 4H). Elemental analyses calc for C_4_H_12_Cl_2_N_2_O_5_Pt: C, 11.07; H, 2.79; N, 6.45. Found: C, 11.35; H, 2.51; N, 6.28.

### 2.2. GO Flake Size Optimization

Commercially available graphene oxide (GO, Sigma Aldrich, Saint Louis, MO, USA) was used as a starting material. The flakes size adjustment and size selection were based on a combination of two previously published protocols [24,25]. Briefly, the GO stock solution (4 mg/mL) was diluted to a concentration of 400 µg/mL in PBS buffer. The diluted GO solution was further sonicated in an ultrasonic bath (Sonorex Digitec DT 103 H, Bandelin, Berlin, ermany) at 70 °C for 6 h. The sample was then agitated for 18 h with a Heidolph (Schwabach, Germany) Unimax 1010 shaker (500 RPM, 65 °C) and sonicated again for 6 h in the ultrasonic bath at 70 °C. Large- flakes were removed by centrifugation (Benchtop 4–16 K, 21191 RCF, 5 min), and the supernatant containing GO dispersion was used in all further experiments.

### 2.3. PEGylation of GO

An amount of 25 mg 8-arm polyethylene glycol-amine (10 kDa, Sigma-Aldrich, St. Louis, MI, USA) was added to the 5 mL of GO dispersion prepared in the previous step and sonicated for 10 min. Subsequently, 40 µL of 5 mg/mL N-(3-Dimethylaminopropyl)-N′-ethylcarbodiimide hydrochloride (EDC, Sigma-Aldrich) was added to the mixture dropwise. Next, a second cycle of agitation and sonification was performed for 18 h (500 RPM, 65 °C) and 6 h (70 °C). The infrared spectra were obtained on a Nicolet iS5 FTIR spectrometer (Fisher Scientific, Waltham, MA, USA) in ATR mode using a ZnSe crystal.

### 2.4. Pt Loading on GO@PEG Nanoplaftorms

Loading of compound **1** onto GO@PEG was carried out in two consecutive steps. A stock solution containing 5 mg of compound **1** and 22.1 mg EDC dissolved in 1 mL deionized H_2_O was resuspended by a brief sonication to form a homogeneous, clear solution, which was agitated (500 RPM) at room temperature for 1 h. A volume of 100 µL of 5 mg/mL compound **1** stock solution was added to 1 mL GO@PEG and agitated for 24 h (500 RPM, 23 °C). Unbound compound **1** was removed by centrifugation at 21,191 RCF for 10 min. The total amount of the anchored compound **1** was determined by atomic absorption spectroscopy (AAS) performed in triplicate (*N* = 3). The resulting pellet GO@PEG-Pt was gently resuspended in 1 mL PBS and stored at 4 °C until use.

### 2.5. Characterization of GO-Based Nanoplatforms 

#### 2.5.1. Determination of GO Amount and Size Distribution

Raman spectra of GO@PEG nanoplatforms were obtained using a Witec Alpha 300 R+ Raman spectroscopic system (Witec, Ulm, Germany) followed by excitation operating at 532 nm. The power of the laser on the sample was 5 mW. In total, 30 microscans were averaged to obtain one spectrum. In total, six spectra from six random locations over the flake were averaged to obtain the data shown in the Section 3. The concentration and size distribution of the prepared dispersion were determined by means of atomic force microscopy (AFM). The concentration was 1.5 × 10^9^ GO flakes/mL, with a median of 266 nm. An atomic force microscope (AFM, Ntegra spectra, NT-MDT, Moscow, Russia) was used to analyse the height and size profile of the GO flakes in the stock solution and in the supernatant solution, as well as GO@PEG (Appendix A). Based on previous AFM observations by many groups, the thickness of single-layer graphene has been experimentally demonstrated to be approximately 1.1 nm [26,27,28]. This microscopic method was also used to determine the amount of GO flakes in 1 mL of supernatant solution and, subsequently, to analyse the size distribution of GO in the supernatant solution.

A volume of 5 μL of the sample was pipetted onto a mica substrate with a radius of 0.5 cm. AFM images of 50 × 50 µm samples were taken in semi-contact mode with an ACTA-SS-10 tip at a scan speed of 0.3 Hz. Subsequently, the captured images were edited in the Gwyddion program, and the number of GO flakes per image was analysed using ImageJ software. The final amount of GO in the supernatant solution was then calculated. The evaluation results in ImageJ were also used for size distribution (*N* = 2142).

Scanning electron microscopy (SEM) images taken using a Hitachi SU6600 scanning electron microscope (Hitachi, Tokyo, Japan). For analysis of GO in stock solution, a small drop of material dispersion in water was placed on a carbon tape and dried at room temperature. An accelerating voltage of 7 kV was used for imaging. A small drop of GO@PEG dispersion in water was placed on a copper grid with carbon film and dried at room temperature. The sample was analysed with an accelerating voltage of 5 kV.

#### 2.5.2. Pt Loading on GO-PEG Nanoplatforms 

The loading efficiency of compound **1** on GO@PEG was determined from the supernatant obtained by centrifugation in the final step of GO@PEG-Pt preparation (Section 2.4) using atomic absorption spectroscopy (AAS). The loading ratio of Pt was analysed by AAS and calculated according to a previously published definition of loading efficiency [28]. LE% is defined as (concentration of the drug loaded on GO/the initial concentration of the drug) × 100:$$\mathrm{LE}=\frac{Concentration\;of\;Pt\;loaded}{concentration\;of\;Pt\;initially}\times 100\%$$

#### 2.5.3. GO-PEG Pt Stability Testing

GO@PEG Pt solution in the dialysis membrane was incubated in a flask with PBS at pH 7.4 for 1, 2, 3, 5, 14 and 21 days under two different temperature conditions (4 °C and 23 °C). The concentration of released Pt from the GO@PEG nanoplatform to PBS was also measured by means of AAS. Measurements were performed in triplicate for each time point (*N* = 3).

### 2.6. In Vitro Biological Study

Compound **1** (Pt-free) was dissolved in DMSO at 1 mg/mLm diluted in the cultured medium at different concentrations (i.e., 15 μM, 30 μM and 60 μM) and used as the Pt-free group. Based on the quantification of compound **1** loaded on the nanoplatforms, the bioactivity of GO@PEG-Pt was tested at the same concentration of Pt-free that corresponded to 1.0 μg/mL, 2.0 μg/mL and 4.0 μg/mL of GO@PEG nanoplatforms loaded with 15 µM, 30 µM and 60 µM of Pt-free, respectively. The GO@PEG nanoplatforms alone were also tested at the same concentration (1.0 μg/mL, 2.0 μg/mL and 4.0 μg/mL) to verify the cytotoxicity of the nanoplatform itself. Cells alone were used as a control group.

#### 2.6.1. Cell Culture

The following cell lines were used: three human osteosarcoma cell lines: MG63 (ATCC CRL1427), SAOS-2 (ATCC HTB-85) and U2-OS (ATCC HTB-96); two human adenocarcinoma cell lines isolated from breast cancers: MDA-MB-231 (ATCC HTB26) and MDA-MB-468 (ATCC HTB231); and two human glioblastoma cell lines: U118 (ATCC HTB15) and U87 (ATCC HTB14). MG63 cells were cultured in a growth medium composed of DMEM F12 GlutaMAX™ modified medium (Gibco, Waltham, MA, USA), 10% foetal bovine serum (FBS) (Gibco) and 1% penicillin/streptomycin (Pen/Strep) (100 U/mL-100 µg/mL, Gibco); SAOS-2 cells were cultured using McCoy’s 5 A (modified) medium (Gibco) supplemented with 15% FBS and 1% Pen/Strep; and U2-OS cells were cultured using McCoy’s 5 A (modified) medium supplemented with 10% FBS and 1% Pen/Strep. MDA MB 321 and 468 cells were cultured in growth media using RPMI 1640 (Gibco), 10% FBS and 1% Pen/Strep; U87 cells were grown in complete medium composed of MEM-α nucleosides no-ascorbic-acid medium (Gibco), 10% FBS and 1% Pen/Strep; and U118 cells were cultured using DMEM high-glucose pyruvate medium (Gibco), 10% FBS and 1% Pen/Strep.

Cells were grown at 37 °C in a 5% CO_2_ atmosphere under controlled humidity conditions, detached from culture flasks by trypsinization and then centrifuged. The cell number and viability were assessed by trypan blue dye exclusion test. All cell-handling procedures were performed in a sterile laminar flow hood.

#### 2.6.2. Cell Viability

The MTT assay is a colorimetric protocol used to quantitatively assess cell viability. 3-(4,5-dimethylthiazol-2—yl)2,5-diphenyltetrazolium bromide (MTT) can be reduced to formazan crystals by metabolically active cells. Briefly, 1.6 × 10^4^ cells/cm^2^ were plated in a 96-well plate. After 24 h, Pt-free, GO@PEG-Pt and GO@PEG at the above-reported concentrations were added to the culture medium and left for 72 h. MTT reagent was resuspended in 1X phosphate-buffered saline (PBS) at a 5 mg/mL final concentration and added to cell culture media in a 1:10 ratio. After 2 h incubation at 37 °C, the solution was discarded, and formazan crystals were dissolved by adding DMSO and shaking for 15 min. The absorbance of three biological replicates (*n* = 3) for each condition was read at 570 nm using a Multiskan FC microplate photometer (Thermo Scientific). The results are represented in graphs with % with respect to cells only. 

#### 2.6.3. Cell Morphology 

Cell morphology of 1.6 × 10^4^ cells/cm^2^ was analysed 72 h after incubation with Pt-free and GO@PEG-Pt at 15 μM, 30 μM and 60 μM concentrations and GO@PEG at the corresponding nanoplatform concentrations. Briefly, cells were washed with PBS 1X and fixed using 4% paraformaldehyde (Merck) for 15 min at room temperature (RT). Membrane permeabilization was performed in PBS 1X with 0.1% (*v*/*v*) Triton X-100 (Sigma Aldrich, Saint Louis, MO, USA) for 5 min at RT. A PBS 1X wash was performed, and then F-actin filaments were highlighted with rhodamine phalloidin (Actin Red 555 Ready Probes™ Reagent, Invitrogen, Waltham, MA, USA), following the manufacturer’s indications for 30 min at RT to visualize the cytoskeletal conformation. DAPI (600 nM) counterstaining was performed for identification of cell nuclei, following the manufacturer’s instructions. Images were captured with an inverted Ti-E fluorescence microscope (Nikon, Tokyo, Japan). A single biological replicate was performed for each condition (*n* = 1).

#### 2.6.4. Quantification of Cellular Uptake of Pt

Inductively coupled plasma optical emission spectrometry (ICP-OES, Agilent Technologies 5100 ICP-OES, Santa Clara, CA, USA) was used to evaluate the platinum drug **1** cell internalization (Pt-free and GO@PEG-Pt). Briefly, 1.6 × 10^4^ cell/cm^2^ were seeded in a 6-well plate for each cell line. After 24 h, Pt-free and GO@PEG-Pt were added at a concentration of 30 μM and incubated for 4 and 24 h, respectively. At each time point, cells were washed with PBS 1X, trypsinised and scraped, reaching a final volume of 400 µL for each sample. Cells were counted using a trypan blue dye exclusion test. ICP-OES was used for quantitative determination of Pt ions. Briefly, the samples were dissolved with 500 µL nitric acid (65 wt.%) and 2.1 mL of Milli-Q water, followed by sonication for 30 min in an ultrasonicated bath. Cells alone were prepared similarly. The analytical wavelength of Pt was 265.945 nm. Pt per cell was quantified, and biological analysis was performed in triplicate for each condition (*n* = 3).

#### 2.6.5. Migration Assay

Cell migration ability was analysed by applying an optimized model of the scratch assay [29]. All the cell lines were seeded in a 24-well plate at a density of 50 × 10^3^ cells/cm^2^. After 24 h, the cell monolayer was scraped in a straight line to create a “scratch” with a p200 pipet tip; then, cells were washed with the same cell medium supplemented with only 2% of FBS to remove cell debris, and Pt-free and GO@PEG-Pt were added at a 30 μM concentration. A first image of the scratch was acquired at time 0, then after 24, 48 and 72 h by an inverted Ti-E Fluorescent microscope. For each acquired image, six measures of scratch width were obtained and analysed quantitatively using ImageJ software. In addition, at times 0 and 72 h, cells were fixed with 4% *w*/*v* paraformaldehyde (PFA), cell nuclei were highlighted by DAPI staining and images were acquired using an inverted Ti-e fluorescent microscope. A biological duplicate was performed for each condition (*N* = 2). 

#### 2.6.6. Statistical Analyses

Results of MTT assays are reported as percentage (%) with respect to cells only ± standard error of the mean (SEM), and data were analysed by two-way analysis of variance (two-way ANOVA) and Tukey’s multiple comparisons test. ICP-OES results are elaborated as picograms of Pt per cell, reported in graphs as mean ± SEM and analysed by two-way ANOVA and Sidak’s multiple comparisons test. Scratch assay results are graphically represented as distance covered (µm) by cells over time towards the centre of the performed scratch, expressed as mean ± SEM plotted on the graphs and analysed by two-way ANOVA and Tukey’s multiple comparisons test. Statistical analysis was performed with GraphPad Prism software (version 8.0.1).

## 3. Results and Discussion

In this work, we focused on the development of GO-based nanoplatforms with high performance in delivering platinum-based drugs in order to overcome the current limitations of clinically used chemotherapeutics, including carboplatin, oxaliplatin and cisplatin, which, once in the human body, are rapidly degraded, with considerable toxicity to off-target organs [23,30].

Among various cancer types, we selected three tumours based on their incidence (breast cancer) and aggressiveness (osteosarcoma and glioblastoma). Breast cancer is the leading cause of death in women, with an incidence of 2.3 million cases worldwide in 2020 [11]. Breast cancer often metastasizes in the liver, lungs, brain and, in 70% of cases, bones [31]. Osteosarcoma is the most common bone cancer affecting young patients, with a poor response to chemotherapy, with a negative impact on patients’ life expectancy [12]. Glioblastoma is a very aggressive type of cancer of the central nervous system generated from the glial cells, with a poor prognosis of life expectancy, with 5-year survival rates of 5 % without any significant improvements in recent decades [21]. A distinctive feature of this work is the wide in vitro testing performed in seven cancer cell lines of selected tumours, where cell viability and morphology were first evaluated with three different concentrations of GO@PEG-Pt. The most promising group was investigated further to evaluate the cellular uptake of GO@PEG-Pt, as it is well-known that Pt becomes activated only once it enters the cells. In addition, our in vitro study focused on another extremely important aspect of developing an anticancer strategy: the possibility of inhibiting cancer cell migration in order to reduce the infiltration of the tumour surrounding parenchyma, which often metastasize throughout the body.

### 3.1. Characterization of GO-Based Nanoplatforms 

The mean flake size of the graphene oxide particles found in the GO stock solution was considerably decreased by a combination of protocols previously described by Ma and Chen [24,25] (see Section 2). The treated GO and GO functionalised using PEG were consequently characterized by AFM (Figure 3A,B). Statistical analysis of the obtained image data confirmed the successful preparation of pristine GO flakes with a mean lateral size of 130 nm for more than 85% of the identified flakes (*n* = 2142) (Figure 3C). A full histogram of the flake size distribution is shown in Appendix A. Modification of the surface with PEG resulted in a change in the height of GO flakes; in the stock solution, the typical height was approximately 1.1 nm (1–2 layers, Appendix A), and after surface modification, the height increased by approximately nine times to 9 nm (corresponding to fewer than 20 layers; Figure 3B). The Raman spectrum in Figure 3D shows dominant D and G bands characteristic of sp2 and sp3 hybridization of carbon containing groups in the nanomaterial. The D/G ratio was 0.75, which indicates several defects in the lattice caused by the present functional groups. GO morphology was characterized by the means of scanning electron microscopy (SEM); the resulting micrograph is shown in Figure 3E.

Despite their hydrophilic nature, the prepared GO flakes rapidly aggregated in the presence of salts and serum components, as observed by Wang and Loutfy [32,33,34]. Therefore, the branched eight-arm PEG-NH_2_ polymer was covalently conjugated to the as-prepared GO to increase its stability under physiological conditions. The successful PEGylation of GO flakes was confirmed by data acquired using infrared spectroscopy (Figure 3F), with prominent bands characteristic of PEG observed. Furthermore, PEG-coated GO flakes were imaged by SEM (Figure 3G). The NH_2_ groups of the GO@PEG were covalently linked to the carboxylic part of Pt-based drug 1, as shown in Figure 2. The resulting GO@PEG-Pt showed high temperature-independent stability in PBS buffer (pH 7.4) solution (Figure 3H). After 24 h, 35% of Pt was released from the GO@PEG nanoplatform before reaching a plateau. A loading efficiency (LE) of 64% was determined for the GO@PEG nanoplatform; this higher LE based on GO is a major advantage over conventional nanocarriers, such as liposomes [35] and solid lipid nanocarriers [36]. The use of eight-arm PEG in our study allowed us to achieve a significantly higher LE of cisplatin (LE = 64%) compared to previous studies, wherein linear PEG (LE = 4.5%) [37] and six-arm PEG (LE = 11%) [38] were used.

In general, GO-based nanoplatforms enable the delivery of higher concentrations of drugs to the tumour region. In the case of drug loading for liposomes and solid lipid nanoparticles, the drug needs to be dissolved or added within the matrix. However, due to the limited solubility of hydrophobic drugs in these matrices, the loading capacities are generally lower than those observed for GO. There is also a significant loss of drug during the synthesis of liposomes and solid lipids [32]. For example, Zhou et al. developed a drug delivery platform based on microsomes composed of poly(2-methacryloyloxyethyl phosphorylcholine)-b-poly(methacrylic acid) copolymer. This system was loaded with cis-diamminedichloroplatinum and used for treatment of osteosarcoma cells. The reported drug loading content was 13.7% [33]. Son K.D. et al. described a drug delivery platform based on calcium-phosphate nanocomposites and evaluated its performance in the delivery of cisplatin, caffeic acid and chlorogenic acid for treatment of osteosarcoma. The drug loading content was between 1% (caffeic acid) and 1.7% (cisplatin) [34]. Li et al. described cisplatin-loaded poly(L-glutamic acid)-g-methoxy poly(ethylene glycol) nanoparticles with an average size of around 43 nm for treatment of osteosarcoma [35].

However, as previously mentioned, the use of GO-based nanoplatforms has its limitations. PEGylation, for which both linear and branched PEG can be used, is an effective strategy to overcome these shortcomings. However, linear PEGylation has several limitations, such as lack of targeting ability due to collapse in the bloodstream and low efficiency in drug loading [39,40,41,42]. Compared to linear PEGs, branched PEGs have more modifiable end groups, which can be used to prepare multifunctional systems with different molecules, including anticancer drugs, fluorescent molecules and targeted ligands. Branched PEGs have better targeting ability, environmental responsiveness and blood circulation, improving drug solubility and bioavailability [43].

### 3.2. Synthesis of the Platinum-Based Prodrug

Complex **1** is a Pt(IV) compound based on a cisplatin scaffold with succinic acid in an axial position that allow for the conjugation to GO functionalized with PEG-NH_2_ groups via coupling with EDC (see Section 2 for details). Pt(IV) species are considered prodrugs because they are intracellularly activated by reduction. It is well known that Pt(IV) species are reduced in the cellular environment (by reducing agents such as glutathione or ascorbic acid), releasing the active Pt(II) scaffold (in this case cisplatin). Using this strategy, cisplatin covalently bound to GO is released upon cellular internalization [44]. 

### 3.3. In Vitro Evaluation of GO@PEG-Pt Bioactivity 

The cytotoxicity of the unloaded GO@PEG nanoplatform was investigated at three different concentrations (i.e., 1.0 μg/mL, 2.0 μg/mL and 4.0 μg/mL) corresponding to the concentrations of GO@PEG nanoplatforms loaded with 15 µM, 30 µM and 60 µM of Pt-free, respectively (see Section 2). Unloaded GO@PEG did not show any significant toxicity, given about 100% of viable cells (Figure 4, Figure 5, Figure 6 and Appendix A), without negatively affecting the cell morphology, visualized by actin filament staining, (Figure 4, Figure 5, Figure 6 and Appendix A) in all the seven cell lines tested. These results confirm that the GO@PEG-based nanoplatform is a promising nanodelivery system, in accordance with previous works showing the absence of toxicity strictly related to GO@PEG, even in vivo with higher concentrations [45,46]; therefore, GO@PEG nanoplatforms have the potential to be administered in vivo with no side effects.

The biological bioactivity of GO@PEG-Pt was then evaluated to validate the nanoplatform as a highly performant nanodelivery system. We investigated the cell viability, morphology, nanoplatform uptake and cell migration in different cancer cell lines compared to Pt-free at three different concentrations (15 μM, 30 μM and 60 μM) and cells alone as a control group (i.e., untreated cells). The data show an evident dose-dependent reduction in the cell metabolic activity in both cases (GO@PEG-Pt and Pt-free) compared to the cells alone in all tested cancer cell lines (Figure 4, Figure 5 and Figure 6). These overall results demonstrate that Pt maintained its action when loaded on GO@PEG nanoplatforms.

A detailed review of the data indicates that osteosarcoma is the most affected tumour, with cell viability reduced to 90% compared to cells alone (Figure 4A–C), confirming the literature on osteosarcoma chemotherapy, which reports that cisplatin is a key and widely used drug [47,48]. The most promising result is related to the significant decrease in cell viability in the GO@PEG-Pt group compared to the Pt-free group, which validates the GO@PEG nanoplatforms as promising Pt vehicles for osteosarcoma treatment at a concentration of 30 µM [49,50] (Figure 4A–C).

In the MG63 cell line, this effect is also clearly visible (*p*-value ≤ 0.001) at the lowest tested Pt concentration (i.e., 15 µM) and, although less pronounced, was also detected in the U118 glioblastoma and the MDA-MB-468 breast cancer cell lines (Figure 5A,B and Figure 6A,B).

A qualitative cell morphology analysis was performed, confirming the cell viability data. First, the cell density is lower in the GO@PEG-Pt group compared to the Pt-free group in the osteosarcoma cell lines, breast cancer cell lines and glioblastoma U118 cell line (Figure 4D, Figure 5C, Figure 6C and Appendix A). MG63 and SAOS-2 cell lines are most negatively affected by the presence of GO@PEG-Pt, with a round and smaller cell morphology shape and with actin filaments aggregated at the cell’s edges (Figure 4D). This analysis confirmed the absence of cytotoxicity of the GO@PEG nanoplatform in all the tested cells (Figure 4D, Figure 5C and Figure 6C). In fact, the cell morphology reflected the healthy state of the cells in the GO@PEG group, showing a high cell number and typical spindle-shaped cells without differences with the untreated cells-only group (Figure 4D, Figure 5C and Figure 6C).

To better elucidate whether the higher cell mortality was strictly related to enhanced Pt uptake driven by the GO@PEG nanoplatform, a cellular uptake analysis was performed and evaluated with ICP-OES, quantifying the Pt amount per cell after 4 and 24 h at 30 µM, selected as the most promising concentration. The GO@PEG nanoplatform enhanced Pt internalization in all the cell lines tested after 24 h, with the greatest differences compared to Pt-free in osteosarcoma and glioblastoma cell lines (MG63 *p*-value ≤ 0.0001; SAOS-2 and U87 *p*-value ≤ 0.05) (Figure 7A–C, Figure 8A,B and Figure 9A,B). However, even after 4 h, a visible trend of the increase in Pt uptake was observed. An evident discrepancy between Pt-free and GO@PEG was observed for osteosarcoma cell lines (*p*-value ≤ 0.0001 and *p*-value ≤ 0.001 in MG63 and U2-OS cells, respectively), glioblastoma cell lines and the MDA-MB-468 breast cancer cell line (Figure 7A–C, Figure 8A,B and Figure 9A,B). 

These results are very promising and support our initial hypothesis with respect to the use of the GO@PEG nanoplatform as a highly performant nanodelivery system for platinum-based drugs. This strategy was demonstrated to be very efficient in reducing the amount of Pt needed in cancer therapy and, consequently, in diminishing the well-known side effects related to Pt-based drugs.

A further investigation was performed to verify whether the proposed GO@PEG nanoplatform delivery system could also play a key role in the inhibition of the cell migration/invasiveness. One clinically distinctive trait of several tumours is the extensive infiltration by cancer cells of the tumour surrounding parenchyma, which often metastasize throughout the body [51]. An in vitro scratch assay was used as an easy, low-cost and well-developed method to measure cell migration of all seven cell lines in contact with GO@PEG-Pt compared to Pt-free at 30 µM for 72 h (Figure 7D–G, Figure 8C–E and Figure 9C–E). The distance covered by cells towards the centre of the scratch was quantified as an indicator of migrating movements, and overall, the results showed that GO@PEG-Pt significantly inhibits the migration of all tested cell lines compared to the control group, with the exception of U87 glioblastoma cells, for which a significant inhibition of cell migration was observed in the presence of the nanoplatform only after 48 h (Figure 7D–F, Figure 8C,D and Figure 9C,D). It is important to note that GO@PEG-Pt was able to significantly reduce the migration of MDA-MB-231 breast cancer cells compared to Pt-free (*p* value ≤ 0.0001 at 48 and 72 h; Figure 9C,E). In particular, at 72 h, a distance of only 141 µm was covered by MDA-MB-231 cells in the presence of the loaded nanoplatform compared to the more than doubled migration distance showed by the cells cultured with Pt-free (i.e., a distance of 300 µm covered). These are very interesting and promising results, as although the MDA-MB-231 cell line is not particularly sensitive to the GO@PEG system in terms of cell viability inhibition, as verified by MTT test, the proposed GO-based nanoplatforms could be used as an effective therapy to reduce tumour metastatic invasion, which is the primary cause of patient mortality during breast cancer progression [52]. MDA-MB-231 is a highly invasive breast carcinoma cell line compared to MDA-MB-468 and is commonly used to model late-stage breast cancer. It is invasive in vitro and, when implanted orthotopically, spontaneously metastasizes to lymph nodes [53,54]. A slight difference in migration was also observed in U118 glioblastoma cells, as well as MG63 and U2-OS osteosarcoma cell lines, when cultured with GO@PEG-Pt compared to Pt-free (*p*-value ≤ 0.0001 at 48 h and *p*-value ≤ 0.01 at 72 h for U118; *p*-value ≤ 0.05 and *p*-value ≤ 0.0001 at 48 and 72 h, respectively, for MG63; *p*-value ≤ 0.0001 and *p*-value ≤ 0.05 at 24 and 48 h, and 72 h, respectively, for U2-OS) (Figure 7D,E,G and Figure 8D,E). U118, MG63 and U2-OS cells were able to cover about 360 µm, 112 µm and 137 µm, respectively, when cultured with GO@PEG-Pt compared to Pt-free (about 411 µm and 167 µm for U118, MG63 and U2-OS, respectively). However, the invasive nature of glioblastoma cells is shown by the scratch test, as well as the malignancy of U118 and U87 strictly related to the high proliferative ability of cells, which makes them difficult to treat, as confirmed by our cell viability results [55]. An increasing number of scientific reports are still discussing the critical relationship between the proliferation and migration of glioblastoma cells in relation to platinum-based nanoplatforms, depending on genetically and morphological different cell lines and various platinum sources affecting these two cellular properties. In this case, the results suggest a increased effect of the GO@PEG-Pt nanoplatform on the migration of U118 cells compared to U87. Moreover, the slight but significant inhibition of MG63 and U2-OS migration confirmed the aggressiveness and metastatic potential of osteosarcoma cells relative to conventional chemotherapeutic drugs, including high-dose platinum-based drugs [56,57,58]. On these bases, our results are very exciting overall, suggesting an improvement of Pt drug action in the migration inhibition of osteosarcoma, especially in MG63, U2-OS and glioblastoma cells when loaded on the GO-based nanoplatform. We hypothesize that this behaviour could be attributed to the cell metabolism GO, which decreases the electron transfer chain activity, limiting ATP production and compromising the assembly of actin filaments fundamental to cell migration and tumour invasiveness [59,60]. 

## 4. Conclusions

Several nanomaterials, such as liposomes, polymeric nanoparticles, metal or carbon nanostructures, are useful tools to selectively target tumour cells, implementing the drug’s pharmacokinetics, increasing the anticancer effect and diminishing the toxic effect at the same time. Their large surface area makes carbon-based nanomaterials excellent drug carrier candidates. In this study, we validated a GO-based nanoplatform modified using eight-arm PEG to improve functionalisation with a Pt(IV) anticancer drug based on a cisplatin scaffold, obtaining a highly performant nanodelivery system as a versatile system in cancer therapy. The obtained data demonstrate that the use of a GO@PEG carrier permits the use of less Pt drugs, achieving a very good cellular proliferation inhibition in osteosarcoma strictly related to a higher cellular uptake. This enhanced cellular internalization is also observed in glioblastoma; however, due to a different cell metabolism, the Pt drug bioactivity, once inside the cells, is less pronounced, although it is a good starting point to drive more selective drugs (e.g., triazene analogue of dacarbazine). The proposed GO@PEG nanoplatform is also promising for the inhibition of migration, especially in highly invasive breast carcinoma (i.e., MDA-MB-231 cell line), neutralizing the metastatic process. In conclusion the GO@PEG nanoplatform represents a promising tool in nanomedicine, in particular for cancer treatment, due to its drug type and loading capacity, which can be specifically tailored to target different cancers. 

## Figures and Tables

**Figure 1 nanomaterials-12-02372-f001:**
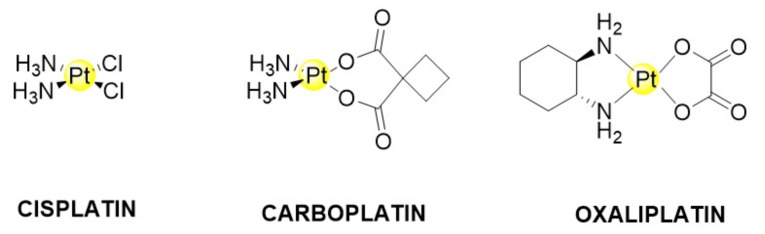
Pt-based drugs. Cisplatin, carboplatin and oxaliplatin.

**Figure 2 nanomaterials-12-02372-f002:**
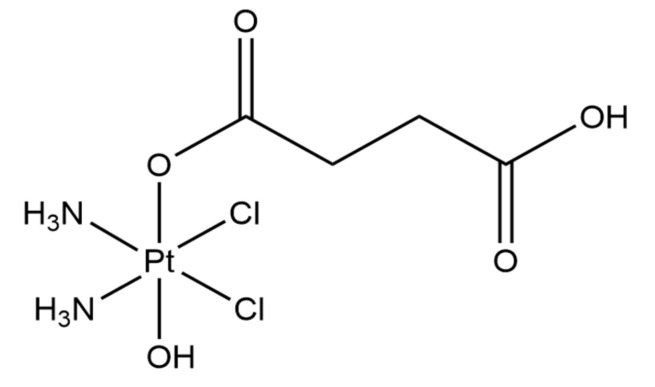
Chemical structure of Compound **1** (Pt-free).

**Figure 3 nanomaterials-12-02372-f003:**
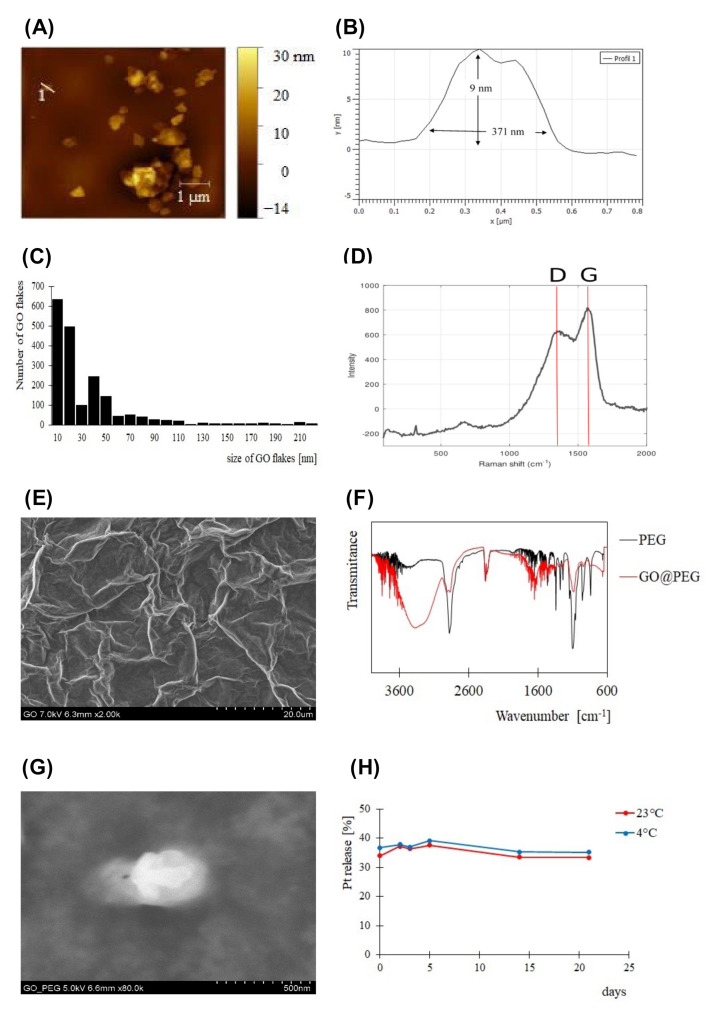
Characterization of GO-based nanoplatforms. (**A**) AFM image of PEGylated GO flakes. (**B**) Height profile was determined for marked GO flakes. (**C**) Size distribution of GO flakes in supernatant and typical D and G bands of GO are shown by the Raman spectrum (**D**). (**E**) Scanning electron microscopy images (SEM) of graphene oxide in stock solution and the presence of PEG in the sample with GO are demonstrated by IR spectra (**F**). (**H**) Successfully PEGylated GO flakes are displayed by SEM (**G**). Release kinetics of Pt-loaded GO@PEG at 4 °C and 23 °C (*n* = 3).

**Figure 4 nanomaterials-12-02372-f004:**
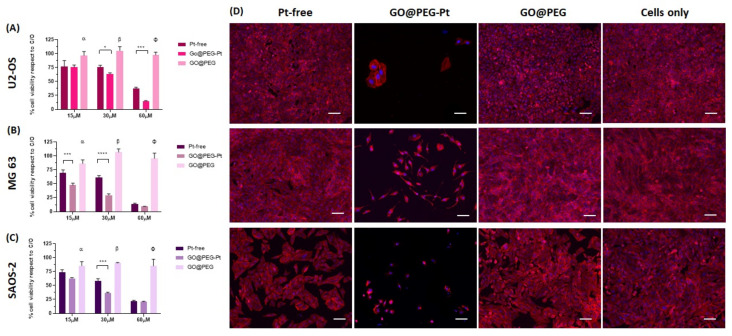
Cell viability and morphological analysis in osteosarcoma cell lines. An MTT assay was performed after 72 h of cell culture. The data show the percentage of viable cells compared to cells alone as the control, and the mean ± standard error of the mean is presented. The graphs show the viability of the U2-OS cell line (**A**), the MG63 cell line (**B**), the SAOS-2 cell line (**C**) and the respective morphological images of the cells cultured for 72 h in the presence of the 30 mM concentration. Phalloidin red stains for actin filaments, and DAPI blue stains for cell nuclei (**D**). Scale bars: 100 µm. (* *p*-value ≤ 0.05; *** *p*-value ≤ 0.001; **** *p*-value ≤ 0.0001). Significant differences between GO@PEG and the other compounds are reported in the graph as follows: a: GO@PEG vs. Pt-free; *p*-value ≤ 0.01 and GO@PEG vs. GO@PEG-Pt *p*-value ≤ 0.001 in U2-OS; GO@PEG vs. Pt-free *p*-value ≤ 0.01 and GO@PEG vs GO@PEG-Pt *p*-value ≤ 0.0001 in MG63; GO@PEG vs. Pt-free *p*-value ≤ 0.05 and GO@PEG vs GO@PEG-Pt *p*-value ≤ 0.001 in SAOS-2. b and F: GO@PEG vs Pt-free and GO@PEG vs. GO@PEG-Pt both *p*-value ≤ 0.0001 in all cell lines.

**Figure 5 nanomaterials-12-02372-f005:**
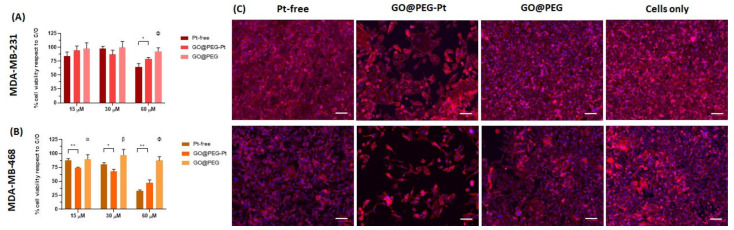
Cell viability and morphological analysis in glioblastoma cell lines. An MTT assay was performed after 72 h of cell culture. The data show the percentage of viable cells compared to cells alone as the control, and the mean ± standard error of the mean is presented. The graphs show the viability of the U87 cell line (**A**), the U118 cell line (**B**) and the respective morphological images of the cells cultured for 72 h in the presence of the 30 mM concentration. Phalloidin red stains for actin filaments, and DAPI blue stains for cell nuclei (**C**). Scale bars: 100 µm. (* *p*-value ≤ 0.05; ** *p*-value ≤ 0.01). Significant differences between GO@PEG and the other compounds are reported in the graph as follows, a: GO@PEG vs. Pt-free *p*-value ≤ 0.01 in U87; GO@PEG vs. GO@PEG-Pt *p*-value ≤ 0.05 in U118. b: GO@PEG vs. Pt-free *p*-value ≤ 0.0001 and GO@PEG vs. GO@PEG-Pt *p*-value ≤ 0.01 in U87, GO@PEG vs. Pt-free *p*-value ≤ 0.01 and GO@PEG vs. GO@PEG-Pt *p*-value ≤ 0.0001 in U118. F: GO@PEG vs. Pt-free *p*-value ≤ 0.0001 and GO@PEG vs. GO@PEG-Pt *p*-value ≤ 0.001 in U87, GO@PEG vs. Pt-free *p*-value ≤ 0.0001 and GO@PEG vs. GO@PEG-Pt *p*-value ≤ 0.0001 in U118.

**Figure 6 nanomaterials-12-02372-f006:**
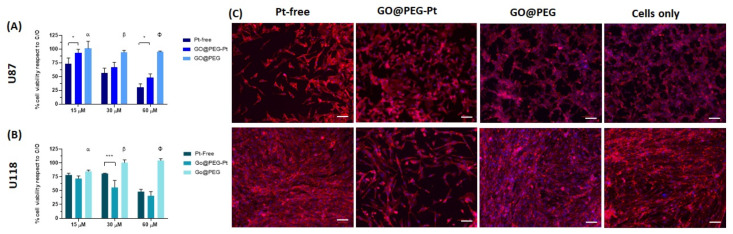
Cell viability and morphological analysis in breast adenocarcinoma cell lines. An MTT assay was performed after 72 h of cell culture. The data show the percentage of viable cells compared to cells alone as the control, and the mean ± standard error of the mean is presented. The graphs show the viability of the MDA-MB-231 cell line (**A**), the MDA-MB 468 cell line (**B**) and the respective morphological images of the cells cultured for 72 h in the presence of the 30 mM concentration. Phalloidin red stains for actin filaments, and DAPI blue stains for cell nuclei (**C**). Scale bars: 100 µm. (* *p*-value ≤ 0.05, *** *p*-value ≤ 0.001). Significant differences between GO@PEG and the other compounds are reported in the graph as follows, a: GO@PEG vs. GO@PEG-Pt *p*-value ≤ 0.05 in MDA-MB 468. b: GO@PEG vs. Pt-free *p*-value ≤ 0.01 and GO@PEG vs. GO@PEG-Pt *p*-value ≤ 0.0001 in MDA-MB 468. F: GO@PEG vs. Pt-free *p*-value ≤ 0.001 in MDA-MB 231, GO@PEG vs. Pt-free and GO@PEG vs. GO@PEG-Pt both *p*-value ≤ 0.0001 in MDA-MB 468.

**Figure 7 nanomaterials-12-02372-f007:**
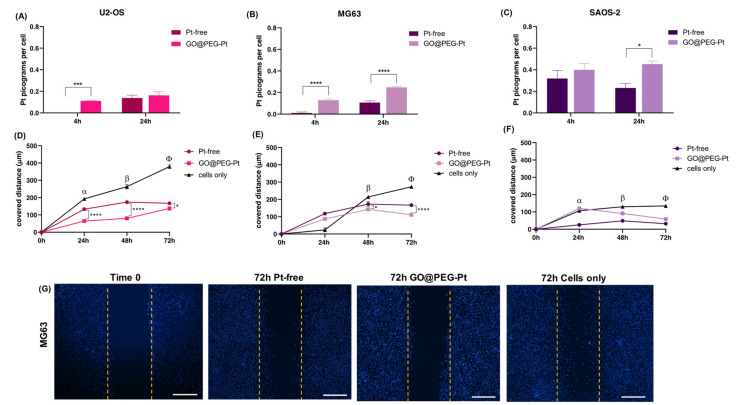
Human osteosarcoma cell lines: ICP-OES on U2-OS (**A**), MG63 (**B**) and SAOS-2 (**C**). Scratch test on U2-OS (**D**), MG63 (**E**) and SAOS-2 (**F**). α: Pt-free vs. cells only and GO@PEG-Pt vs. cells only, both *p* values ≤ 0.0001 in U2-OS; Pt-free vs. cells only. *p* value ≤ 0.0001 in SAOS-2. β: Pt-free vs. cells only and GO@PEG-Pt vs. cells only, both *p* values ≤ 0.0001 in U2-OS; Pt-free vs. cells only, *p* value ≤ 0.01, and GO@PEG-Pt vs. cells only, *p* value ≤ 0.0001 in MG63; Pt-free vs. cells only and GO@PEG-Pt vs. cells only, both *p* values ≤ 0.0001 in SAOS-2. φ: Pt-free vs. cells only and GO@PEG-Pt vs. cells only both, *p* value ≤ 0.0001 in U2-OS, MG63 and SAOS-2. Significant differences between Pt-free and GO@PEG-Pt are reported in the graph as follows: * *p*-value ≤ 0.05, *** *p*-value ≤ 0.001, **** *p*-value ≤ 0.0001. Representative DAPI staining of scratch test on MG63 cells (**G**) cells. Scale bars = 500 µm. Cell nuclei are indicated in blue.

**Figure 8 nanomaterials-12-02372-f008:**
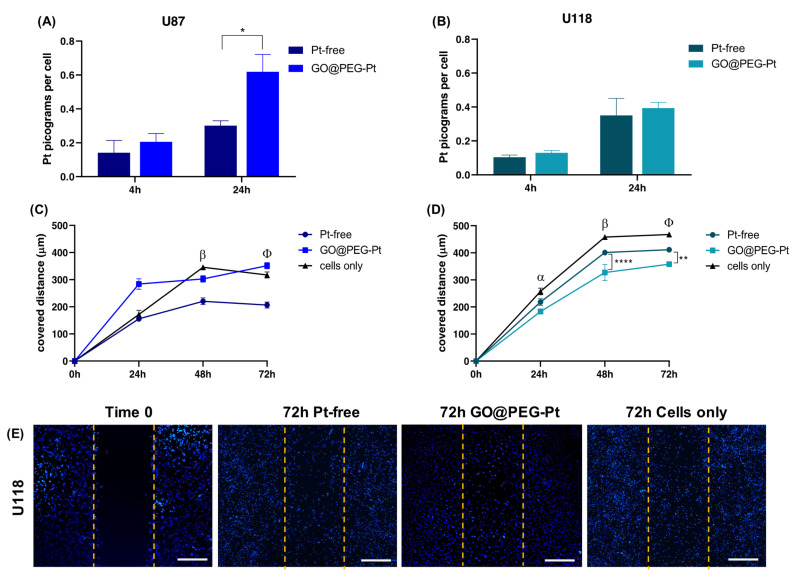
Human glioblastoma cell lines. ICP-OES on U87 (**A**) and U118 (**B**). Scratch test on U87 (**C**) and U118 (**D**); α: Pt-free vs. cells alone, *p* value ≤ 0.05, and GO@PEG-Pt vs. cells alone, *p* value ≤ 0.0001 for U87. β: Pt-free vs. cells alone, *p* value ≤ 0.0001, and GO@PEG-Pt vs. cells alone, *p* value ≤ 0.05 in U87; Pt-free vs. cells alone, *p* value ≤ 0.001, and GO@PEG-Pt vs. cells alone, *p* value ≤ 0.0001 in U118. φ: Pt-free vs. cells alone, *p* value ≤ 0.0001 in U87; Pt-free vs. cells alone, *p* value ≤ 0.001, and GO@PEG-Pt vs. cells alone, *p* value ≤ 0.0001 in U118. Significant differences between Pt-free and GO@PEG-Pt are reported in the graph as follows: * *p*-value ≤ 0.05, ** *p*-value ≤ 0.01, **** *p*-value ≤ 0.0001. Representative DAPI staining of scratch test on U118 cells (**E**). Scale bars = 500 µm. Cell nuclei are indicated in blue.

**Figure 9 nanomaterials-12-02372-f009:**
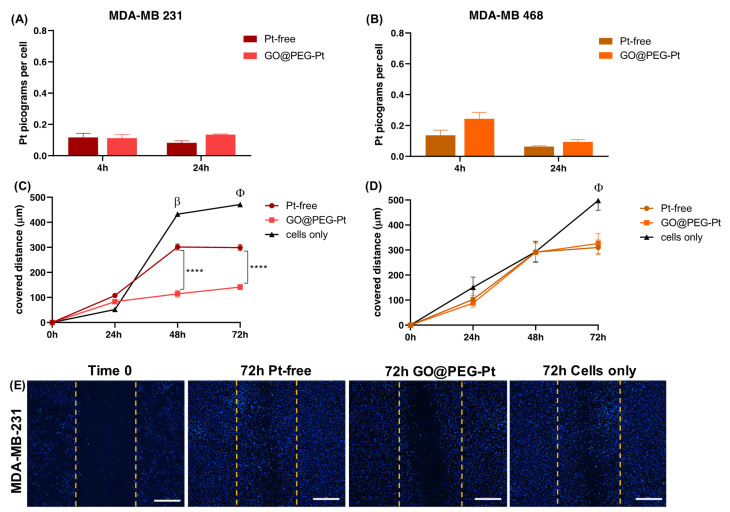
Human breast adenocarcinoma cell lines. ICP-OES on MDA-MB-231 (**A**) and MDA-MB-468 (**B**). Scratch test on MDA-MB-231 (**C**) and MDA-MB-468 (**D**); β: Pt-free vs. cells alone and GO@PEG-Pt vs. cells alone, both *p* values ≤ 0.0001 in MDA-MB 231. φ: Pt-free vs. cells alone and GO@PEG-Pt vs. cells alone, both *p* values ≤ 0.0001 in MDA-MB 231; Pt-free vs. cells alone, *p* value ≤ 0.0001, and GO@PEG-Pt vs. cells alone, *p* value ≤ 0.001 in MDA-MB 468. Significant differences between Pt-free and GO@PEG-Pt are reported in the graph as follows: **** *p*-value ≤ 0.0001. Representative DAPI staining of scratch test on MDA-MB-231 cells (**E**). Scale bars = 500 µm. Cell nuclei are indicated in blue.

## Data Availability

Not applicable.

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
