# Peer review of "Graphene Oxide Nanoplatforms to Enhance Cisplatin-Based Drug Delivery in Anticancer Therapy"

_nanomaterials, 2022, doi:10.3390/nano12142372_

Round 1
Reviewer 1 Report
In this work, the authors provided a method to load cisplatin into GO-PEG. This work showed some good in vitro results, but it is not enough to demonstrate the merits of cisplatin loaded GO-PEG system. This work can not be published in this journal after the authors adding in vivo tests.
1. GO could be degraded or clear out the body? If the GO retains in body, is there some side effects on the body?
2. How do you test the loading rate and release kinetics of cisplatin from GO-PEG? HPLC? you should provide a detailed protocol.
3. you mention the tumor cells in the work, so how do you know GO-PEG could penetrate the brain-blood barrier?
4. I think only in vitro data is not enough to support the conclusion.
Reviewer 2 Report
The work titled "Graphene oxide nanoplatforms to enhance cisplatin-based drug delivery in anticancer therapy" aims to fabricate and characterize a 2D graphene oxide-based nanoplatform functionalized with highly branched 8-arm polyethylene glycol. The proposed GO @PEG nanoplatform also shows promise in inhibiting migration, particularly in highly invasive breast cancers, by neutralizing the metastatic process. How do you compare to published data? What is the added benefit? Details about number of samples, replicate… should be provided. About analysis statistical, how was performed? A resume integrative picture could be done to provide an illustration of data and transpose it.
Please check line 539, it is not perceptible.
Round 2
Reviewer 1 Report
The language needs to be improved, and try to improve the disscussion part, compare you work with the published reports.
Author Response
We revised the manuscript following the reviewer's suggestion.